# Climate Overrides the Influence of Microsite Conditions on Radial Growth of the Tall Multi-Stemmed Shrub *Alnus alnobetula* at Treeline

**DOI:** 10.3390/plants12081708

**Published:** 2023-04-20

**Authors:** Walter Oberhuber, Anna-Lena Dobler, Tamara Heinzle, Francesca Scandurra, Andreas Gruber, Gerhard Wieser

**Affiliations:** Department of Botany, University of Innsbruck, A-6020 Innsbruck, Austriaandreas.gruber@uibk.ac.at (A.G.); gerhard.wieser@uibk.ac.at (G.W.)

**Keywords:** climate forcing, green alder, growth variability, multi-stemmed shrub, radial stem growth, ring width, tree ring

## Abstract

Green alder (*Alnus alnobetula*), a tall multi-stemmed deciduous shrub, is widespread at high elevations in the Central European Alps. Its growth form frequently leads to asymmetric radial growth and anomalous growth ring patterns, making development of representative ring-width series a challenge. In order to assess the variability among radii of one shoot, among shoots belonging to one stock and among stocks, 60 stem discs were sampled at treeline on Mt. Patscherkofel (Tyrol, Austria). Annual increments were measured along 188 radii and analyzed in terms of their variability by applying dendrochronological techniques. Results revealed a high agreement in ring-width variation among radii of one shoot, among shoots of one stock and largely among stocks from different sites, confirming the pronounced limitation of radial stem growth by climate forcing at the alpine treeline. In contrast to this, a high variability in both absolute growth rates and long-term growth trends was found, which we attribute to different microsite conditions and disturbances. These factors also override climate control of radial growth under growth-limiting environmental conditions. Based on our findings we provide recommendations for the number of samples needed to carry out inter- and intra-annual studies of radial growth in this multi-stemmed clonal shrub.

## 1. Introduction

During the last decades, land use change and climate warming favor the spreading of shrubs in arctic and alpine environments (e.g., [1,2,3]), resulting in widespread impacts on ecosystem services, plant diversity, soil nutrient content and biogeochemical cycles [4,5,6,7,8]. The most expanding shrub species in the European Alps is green alder (*Alnus alnobetula* (Ehrh.) K. Koch former *Alnus viridis* (Chaix) DC; [9,10,11]), a tall multi-stemmed early successional shrub forming dense 2–4 m tall canopies with shoots showing arched ascending growth [12,13]. The spreading of tall *Alnus* spp. shrubs is also reported for the tundra ecotone [14,15,16]. Due to clonal growth by layering, stocks are developed from which numerous shoots sprout [17]. Preference for clonal growth over vertical stem growth is part of the successful and rapid expansion strategy within the treeline ecotone triggered primarily by decreasing grazing pressure and land abandonment [18,19,20].

Due to environmental alterations that occur in the course of shrub expansion, the drivers of shrub growth are being increasingly studied by applying dendrochronological techniques (for a review see Myers-Smith et al. [21]). With regard to tall shrubs, dendroclimatic analyses were applied to quantify the climatic sensitivity of radial growth of, e.g., *Salix lanata*, *Salix glauca*, *Alnus viridis* ssp. *fruticosa* and *Alnus alnobetula* [22,23,24,25,26]. These studies focused on the mean climate response of all individuals (=stocks) at a given site, disregarding variability in radial growth among radii of one shoot, shoots belonging to one stock and stocks. In multi-stemmed *Alnus alnobetula*, ascending growth of stems often leads to a pronounced asymmetric radial growth, forming ellipse-shaped growth rings with eccentric pith in the lower ascending stem region, which is enhanced by heavy snow load occurring in avalanche tracks. Furthermore, above the closed forest, environmental conditions and disturbances (e.g., snow load, wind) frequently change at small spatial scales and temperature gradients, resulting in differences in insolation that can develop within tall multi-stemmed shrubs [27]. For this reason, it can be assumed that radii of one shoot and shoots belonging to one individual may show high variability in radial growth due to varying microclimates and disturbances. Hence, the development of reliable ring-width series necessary to determine, e.g., climate forcing of radial growth or long-term growth trends in response to climate warming, is associated with great difficulties due to frequently poor agreement among ring-width series that are (i) measured along different radii, (ii) developed from different shoots belonging to one individual or (iii) among individuals. In order to cope with these challenges, several authors suggested methodological approaches for dendrochronological studies of especially dwarf shrubs [21,28,29,30].

The main focus of this study was to evaluate the variability in radial growth among radii of one shoot, among shoots of one individual and among individuals of the tall multi-stemmed shrub *Alnus alnobetula* spreading above 2100 m asl within the alpine treeline ecotone on Mt. Patscherkofel (Tyrol, Austria; [26]). Based on our considerations, we formulated the following three Hypotheses (H):

**H1.** 
*Radial growth measured along several radii of one shoot shows high variability in absolute growth and low synchrony in year-to-year ring-width fluctuations.*


**H2.** 
*Radial growth compared among radii of different shoots within one individual (stock) shows low variability in absolute growth and high synchrony in year-to-year ring-width fluctuations.*


**H3.** 
*Radial growth compared between different individuals exposed to site-specific environmental conditions and disturbances shows high variability in absolute growth and long-term growth trends and low synchrony in year-to-year ring-width fluctuations.*


Recording growth variability among different radii of one shoot, among shoots belonging to one individual and among individuals will allow specifying how many radii and shoots of one multi-stemmed individual and how many individuals must be sampled in order to develop representative ring-width time series of this tall multi-stemmed shrub. An adequate sampling design is a prerequisite to determine climate forcing of radial stem growth and long-term growth trends in this clonal shrub spreading rapidly at the alpine treeline.

## 2. Results

Collected shoots had a mean stem length of 2.5 m (stocks IM and AG) and 1.6 m (stock WE), and mean stem diameter ranged between 2.7 cm (stock WE) and 4 cm (stock AG; Table 1). Mean cambial age close to the stem base varied between 23 and 29 yrs, and mean ring width was 462 µm (stock WE), 566 µm (stock IM) and 618 µm (stock AG). Recorded growth parameters as well as shoot age of stock WE differed significantly (*p* ≤ 0.01) from the other individuals (i.e., stocks IM and AG).

### 2.1. Variability in Radial Growth among Radii of One Shoot

Ring-width series of all radii (cf. Figure 1) recorded in all shoots taken from one stock are depicted in Figure 2a–c. In all three stocks investigated, radius *a* was significantly wider than radii *b*, *c* and *d* and varied between 508 to 703 µm at the WE and AG sites, respectively (Figure 2d–f). At the AG site, mean ring width of radius *d* was also significantly different (*p* ≤ 0.01) from mean ring widths of radii *a* and *c*. Mean ring widths of opposite radii (=*ab* and *cd*) were significantly different (*p* ≤ 0.05) from each other at the IM and AG sites, but were not significantly different from mean ring widths of all radii (=*abcd*) in all stocks (Figure 3).

The agreement of inter-annual radial growth among radii of one shoot is depicted in Figure 4a–f. The highly significant agreement (*p* ≤ 0.001) in ring width variations of radius *a* and *b* (i.e., *a*:*b*) and radius *a* and mean of radii *c* and *d* (i.e., *a*:*cd*), as well as mean of radii *a* and *b* and *c* and *d* (i.e., *ab*:*cd*), are striking (Figure 4a–c). Mean *t*_BP_-scores for all radius comparisons in all individuals were more than twice as high as the value of 3.5, indicating highly secure cross-dating.

### 2.2. Variability in Radial Growth among Radii of Different Shoots of One Stock

All statistical parameters estimating agreement in year-to-year variation of ring-width series (i.e., EPS, S/N-ratio, EV and Rbar) were high for all single radii, mean values of opposite radii and mean of all radii (Table 2). The highest values for single radii (i.e., radius *a*, *b*, *c* or *d*) were obtained for radius *a* at the WE and AG sites. These statistics increased only slightly when calculated on the basis of the mean value of all radii. MS values were quite similar in all radii (including mean values calculated from different radii) and all stocks, ranging from 36.7% to 48.9%. AC was rather low (*r* < 0.26) in all radii recorded in shoots sampled at the WE and AG sites. AC reached at most a value of *r* = 0.430 at the IM site.

Agreement of inter-annual radial growth among single radii and combinations of radii (=*ab*, *cd* and *abcd*) of shoots belonging to the same stock are depicted in Figure 5a–f. It is evident that radius *a* achieved the highest mean percentage of sign agreement (*Glk*; Figure 5a–c) and mean *t*_BP_-score (Figure 5d–f) of all single radii in all individuals. Mean *Glk* for radius *a* of all shoots belonging to the same stock were statistically significant at *p* ≤ 0.01 (stock WE) and at *p* ≤ 0.001 (stocks IM and AG). Mean *t*_BP_-score for radius *a* was close to (stock WE) or higher than (stocks IM and AG) 3.5. Time series of mean ring width calculated from measurement of all radii (=*abcd*) show high agreement in year-to-year variation in radial growth in shoots taken from one stock (Figure 5a–f and Figure 6a–c). Larger radial growth variability in favorable years (e.g., 2003, 2015, 2017) than in years having poor growing conditions (e.g., 2002, 2004, 2016) are reflected by a wider and narrower interquartile range, respectively. There is also a pronounced coincidence in radial growth increase and reduction in 2003 and 2016, respectively, among shoots and WE and IM individuals. Ring width in 2003 is not significantly different from the mean in individual AG. However, in these extreme growth years, mean ring widths in all individuals were significantly different (*p* ≤ 0.001) from the preceding year. Mean ring widths of all four radii measured (Figure 6d–f; “all years”) are significantly (*p* ≤ 0.001) lower at site WE compared to mean ring widths recorded at other sites, i.e., IM and AG, which are not significantly different (*p* = 0.062).

### 2.3. Variability in Radial Growth among Different Sites

Agreement among ring-width series (single radius and combinations of radii) of different individuals are given in Table 3. Stocks WE and IM show high *Glk* (*p* ≤ 0.001) and *t*_BP_-scores (close to or higher than 7) for all single radii and calculated means (=*ab*, *cd*, *abcd*). *Glk* values are also significant (*p* ≤ 0.05) for all radii among ring-width series at sites WE and AG (mean *t*_BP_-score = 2.4). The lowest level of agreement was found among ring-width series of single radii of stock IM and AG (mean *t*_BP_-score = 3.6). Radial growth of combinations of radii (i.e., *ab*, *cd*, *abcd*) are statistically significantly different (*p* ≤ 0.05) between stock WE and stocks IM and AG, but not between the latter two (Figure 3). Although trends in ring-width series at different sites show a general consistency, they differ strongly with respect to the more recent growth trend. While at site AG there is a decreasing growth trend, constant growth is detectable at site WE, and at site IM *Alnus alnobetula* shows the most steadily increasing growth after the low-growth period occurring around 2007 (Figure 7).

## 3. Discussion

The multi-stemmed shrub *Alnus alnobetula* shows clonal growth by layering and root sprouts (e.g., [31]). The vegetative horizontal spread can be triggered by mechanical disturbances, e.g., avalanches, high snow load, wind exposure or livestock grazing. Shoots from one rootstock therefore emerge at intervals and have varying ages (Table 4). The highest variability in cambial age, ranging from 19 to 35 years, was found for shoots sampled in the avalanche track (site AG), indicating that this stock was exposed to the most frequent occurrences of disturbances, which most likely have induced successive sprouting. Extreme environmental conditions are most likely responsible for the significantly lower elongation and diameter growth at the site WE compared to other stocks (Table 4). To be specific, shallow soil depth and strong wind exposure on the south-facing ridge may impair water availability and make shoots of the individual WE also highly susceptible to frost drought during winter [26].

Low autocorrelation in ring-width series of all radii and radius combinations (Table 2) indicates that current climate conditions primarily determine radial growth of this deciduous shrub, which is consistent with a conceptual tree model put forward by Zweifel and Sterck [32] that shows that legacy effects are related to lifetimes of organs and reserves.

### 3.1. Variability in Radial Growth and Synchronicity among Radii of One Shoot

We expected that due to strong bending of shoots causing asymmetric radial growth and eccentric pith, mean ring width would differ between several radii. However, we found that in two of three individuals, only mean ring width of the longest radius (radius *a*) significantly differed from mean ring width of all other radii (Figure 2). In stock AG occurring within an avalanche track, growth was significantly different between several radii, which indicates that under a high disturbance regime (i.e., high and long duration of snow cover), cambial activity along different radii is more strongly influenced by irregular tension and compression load. Because radial growth only takes place during the snow-free period [27], we assume that due to heavy and prolonged snow load, shoots do not fully return to their original position, which accounts for the differences in mean ring width among radii.

*Glk* and *t*_BP_-scores among radii belonging to the same shoot of one individual show very high agreement in year-to-year variability of radial growth independent of radius measured (Figure 4) and thus contradict our hypothesis (H1). Our results suggest that although irregular tension and compression load can affect amount of total annual increment (i.e., ring width of radius *a* is significantly wider than that of other radii), year-to-year variability in ring width of all radii is determined by environmental forcing, most likely climate forcing. Climate factors most strongly related to radial growth of *Alnus alnobetula* within the study area are summer temperature and winter precipitation [26]. Hence, results corroborate our recent finding that climate forcing is the primary determinant of cambial activity in *Alnus alnobetula*, irrespective of differences in physical strain acting on different radial directions, e.g., of the upper and lower side of the stem.

High values of MS (primarily > 40%) as well as statistics of chronology homogeneity of all single radii and combinations of radii found at all sites (Table 2) also indicate a strong and homogenous response of cambial activity to environmental forcing. It is worth noting that ring-width series of radius *a* having the widest ring widths in all individuals and also show the predominantly highest homogeneity statistics compared to other single radii (Table 2). This finding supports the use of the fastest growing radius, mainly occurring on the lower side of the stem, for recording intra-annual stem radius variations by applying point dendrometers. Averaging four radial measurements in the crosswise direction increases statistics of chronology homogeneity compared to radius *a* only slightly. Somewhat lower but still acceptable statistics were found in the individual located in the avalanche track (site AG), indicating that at a high disturbance regime, a decrease in agreement of inter-annual radial growth occurs among samples (here, shoots belonging to the same stock). This finding is well-known in dendrochronological studies by taking care of tree and site selection, i.e., avoidance of undesirable disturbance factors as a principle of sampling (e.g., [33,34]).

### 3.2. Variability in Radial Growth and Synchronicity among Shoots Belonging to One Stock

A high level of agreement among ring-width time series (single and combinations of radii) of shoots belonging to the same stock (Figure 5a–f) indicates strong climate forcing of radial growth, confirming our hypothesis (H2). Ring-width series of the longest radius (=radius *a*) achieve the highest agreement (synchronicity) among shoots in all individuals. That the growth rate affects agreement in year-to-year growth variability is also supported by higher values of *Glk* and *t*_BP_-scores at sites IM and AG, showing significantly wider annual increments compared to stock WE. We suggest that under generally growth limiting environmental conditions prevailing at site WE (Table 4), disturbances limited to individual shoots can override climate forcing of radial growth, especially in “average” climate years, leading to lower synchrony in ring-width series.

Although based on an average of four radii measured per stem disc, high variability in radial growth among shoots of one individual shows itself through a large interquartile range, which is especially pronounced in favorable years (Figure 6a–f), confirming our hypothesis (H3). Because in *Alnus alnobetula* a closed canopy growth form is not generally developed, growth variability among shoots of one individual might be explained by influence of small-scale differences in microclimate on cambial activity. That a strong influence of canopy cover on microclimate might occur at the treeline has been demonstrated for deciduous beech (*Fagus sylvatica*) by Rita et al. [35].

During the heat wave that occurred in the extraordinary summer of 2003, when mean summer temperatures exceeded the 1961–1990 mean by c. 3 °C in Central Europe [36], a significant increase in radial growth was detected, which corroborates strong temperature forcing of radial growth (cf. Appendix A). In contrast, an exceptional late frost event (daily mean temperatures fell below −10 °C in late April to early May 2016) combined with a cool summer may have resulted in a distinct reduction of radial growth for all individuals in that year (cf. [26]). A significant increase and decrease in radial growth in 2003 and 2016, respectively (*p* ≤ 0.001), compared to the previous year are found in all three stocks independent of site conditions (Figure 6d–f). Results indicate that extreme temperature forcing overlays microclimatic conditions and disturbances, which are present on a small scale and cause high growth variability and partially reduced agreement in year-to-year variability of radial growth at selected sites (Table 3).

The first few ring widths near the pith occasionally show reduced agreement among shoots (Figure 6a–c), and growth trends in ring-width series (even when developed from four radii) can be very pronounced and run in opposite directions in ring-width series developed from shoots belonging to the same individual, i.e., wide rings at low cambial age are becoming narrower with increasing age and vice versa. Most likely, the influence of environmental conditions (e.g., duration of snow cover, wind exposure, frost drought) and microtopography differently affect not only growth of young shoots, which are particularly sensitive to mechanical stress (e.g., snow load), but also growth of shoots over longer periods of time. A change in resource allocation over time between aboveground and belowground growth [37] and among shoots belonging to the same stock [21] has also to be considered.

### 3.3. Variability in Radial Growth and Synchronicity among Stocks

Significantly lower annual increments at site WE compared to sites IM and AG were found (Table 1 and Figure 6d–f). Because the elevational differences among sites are rather small (at most 35 m; Table 4), growth variability can be explained by differences in water and nutrient availability during the growing season. Shallow soil depth and significantly lower soil moisture at site WE (Table 4) compared to other sites supports this assumption. Although the long-term trend in radial growth over the past 30 years is largely consistent among individuals (Figure 7), striking differences in growth rates have occurred in recent years, i.e., growth rates are decreasing at site AG, constant at site WE or increasing at site IM. Changes in disturbance intensity, e.g., duration of snow cover or site-specific impacts of climate warming that affect the growth of single individuals, are likely the cause of temporarily occurring differences in growth trends. Although the basal area increment (BAI) reduces the effects of age and stem size on growth trends and is more closely related to biomass increment than ring width (e.g., [38,39]), BAI could not be adequately determined due to the frequent occurrence of pronounced stem eccentricity. It should also be noted that determining long-term growth trends from tree-ring series is generally a difficult task [40,41].

Highly synchronized radial growth between individuals at sites WE and IM (Table 3) suggests that similar climate variables control radial growth of these individuals. In contrast, inter-annual radial growth among individuals exposed to different aspects shows lower synchronicity, i.e., lower values of statistical parameters (*Glk*, *t*_BP_-scores; Table 3). These findings corroborate results of a previous study within the study area, which required the elimination of about a quarter of the measured ring-width series to create a composite chronology out of eight study plots [26]. A high sensitivity of radial growth to site-specific environmental conditions (MS amounts to *c*. 40%; Table 2), which can change rapidly over small spatial scales within the treeline ecotone [27], is most likely responsible for lower agreement in ring-width variations among some individuals. For example, in the period 2005–2007, stock AG shows opposite growth variations compared to stocks WE and IM occurring on a SSE facing slope (Figure 7), which suggests the modulation of regional climate factors through site-specific conditions, e.g., water or nutrient availability, or disturbances such as avalanches (cf. [15]). Hence, our hypothesis (H3) that radial growth (absolute and trend) and year-to-year variations are influenced by prevailing site-specific environmental conditions is largely confirmed by results of this study.

### 3.4. Recommended Sampling Design for Determining Climate Forcing and Radial Growth Rates of Alnus alnobetula

The high agreement (=synchronous growth) among radii of one shoot and among shoots of one individual, which are also exposed to quite different environmental conditions (Table 4), underline the pronounced limitation of radial growth of *Alnus alnobetula* at treeline by climatic factors. Recently, we found that summer temperature and winter precipitation are the main climate drivers of radial stem growth of this tall shrub within the study area [26]. On the other hand, site-specific environmental conditions (e.g., water and nutrient availability) and/or disturbance factors (e.g., high snow load, wind exposure) cause a decrease in synchronicity among ring-width series developed at different sites ([26]; this study). Therefore, site-specific variability in climate-growth relationships of *Alnus alnobetula* must be considered in topographically heterogenous environments, which led us to the following recommendations: measuring 1 radius per shoot (preferably the one with the longest radius), 3 shoots per individual, and c. 10 individuals within 1 stand, in which individuals are exposed to the same (i.e., homogeneous) environmental conditions (total c. 30 radii per study plot). Note that the number of individuals to be sampled within one plot depends on the variability in radial growth among individuals, which we recommend determining in advance.

However, the determination of absolute growth rates as well as growth trends (e.g., with respect to recent climate warming) is strongly hampered by the pronounced variability in ring width (i) among radii of one shoot, (ii) among shoots of one individual and (iii) among individuals exposed to different environmental conditions and/or disturbance regimes. To be able to take this growth variability into account, the following recommendation is made: measuring 4 radii per shoot (preferably in crosswise-oriented radii along the longest and shortest diameter and excluding stem discs showing extreme abnormal growth), ≥5 shoots per individual, ≥10 individuals within 1 stand, in which individuals are exposed to the same (i.e., homogeneous) environmental conditions (total c. 200 radii per study plot). At sites characterized by a high frequency of disturbances (e.g., in highly active avalanche tracks or grazed areas), a higher number of shoots and individuals is required to account for anomalous growth patterns, discontinuous rings and extreme asymmetric growth.

### 3.5. Recommendations for Determining Intra-Annual Radial Growth

Our results are also relevant for studies focusing on intra-annual dynamics of radial growth of tall shrubs by applying dendrometers, which allow determination of (i) key phenological stages of cambial activity (i.e., onset, end, time of maximum growth and growth duration) and (ii) influence of seasonal and short-term extreme weather conditions on radial growth (e.g., [42,43,44]). Due to pronounced variability in growth rates among shoots sprouting from one individual, ≥5 shoots per individual should be equipped with appropriate measuring devices. Preliminary point dendrometer records gathered from *Alnus alnobetula* within the study area confirmed the high growth variability among shoots found in this study, and also revealed pronounced temporal differences (up to three weeks) in key phenological dates of radial growth, demonstrating the high sensitivity of cambial activity to microclimate in this species. Thus, in future studies we suggest the use of band dendrometers instead of point dendrometers, as the former integrate growth over the entire stem circumference. On the other hand, we suggest that point dendrometers be mounted on the radius showing the highest growth rate (usually at the lower side of the stem), because inter-annual agreement in radial growth is related to growth rate.

## 4. Materials and Methods

### 4.1. Study Area and Site Selection

The study area is within the treeline ecotone of Mt. Patscherkofel (2246 m asl) in the Central European Alps (Tyrol, Austria; 47°12′ N, 11°27′ E) where *Alnus alnobetula* stands are developed between c. 1950 and 2200 m asl primarily in leeward avalanche tracks but also on south- to southeast-facing wind-exposed slopes. The geology of the study area is dominated by gneisses and schists [45]. According to the World Base for Soil Resources [46], the soil type within the study area is classified as haplic podzol [47]. The local climate is characterized by the frequent occurrence of strong southerly foehn winds [48]. During the period 1991–2020, the mean annual temperature was 0.8 ± 0.7 °C and the coldest and warmest months were February (−6.6 °C) and July (8.9 °C), respectively (Appendix A). The mean annual precipitation during this time period was 889 mm, with the majority falling during summer (June–August; 371 mm), while the winter (December–February) was the driest season (134 mm). Snow depth shows pronounced spatial variation due to irregular distribution caused by strong winds. While at south-facing wind-exposed slopes snow depth hardly is >1m and only small patches of snow exist till early April, on leeward north-facing slopes (particularly in avalanche tracks) snow height can reach up to 3 m and a permanent snow cover may persist till late spring and into early June.

### 4.2. Sampling and Growth Ring Measurements

In order to cover a large range of environmental conditions, we selected three *Alnus alnobetula* stocks exposed to different site conditions and disturbance regimes between 2115 to 2150 m asl (Table 4, Appendix A): a north-facing avalanche gully (hereafter referred to as the AG site or AG stock), a wind-exposed ridge (hereafter WE site or WE stock) and an intermediate site (hereafter IM site or IM stock). At each site, soil moisture at 10 cm soil depth was recorded every 30 min during July and August 2022 using four ThetaProbe ML2 devices (Delta-T Devices Ltd., Burwell, UK). Snow load and water availability were at maximum and minimum at the AG site and at the WE site, respectively, and were intermediate at the IM site (Table 4). Accordingly, stem bending and growth were highest at the AG site and lowest at the WE site.

**Table 4 plants-12-01708-t004:** Description of selected study sites (WE = wind-exposed ridge; IM = intermediate site; AG = avalanche gully; asl = above sea level). Statistically significant differences of mean values among sites are indicated by different letters (*p* ≤ 0.01; Student’s *t*-test).

Site	Elevation (m asl)	Aspect	Topography	Slope (°)	Soil Depth (cm)	Soil Moisture (Vol.%)
WE	2150	SSE	Ridge	5	5–10	11.5 ± 3.0 ^a^
IM	2140	SSE	Slope	10	10–15	20.1 ± 2.7 ^b^
AG	2115	N	Avalanche gully	20	NA ^1^	23.0 ± 3.2 ^c^

^1^ Boulder debris impaired the correct determination of soil depth.

Special care was taken to ensure that selected shoots belonged to one stock, i.e., one individual, and that no vegetative links existed to other stocks nearby (cf. [31]). Around the rootstock, i.e., from all cardinal directions, we sampled 20 shoots from each stock (i.e., about half of all shoots sprouting from one stock). All shoots of one stock were taken within a radius of no more than 50 cm from the center. Because the lifespan of shoots is less than 50 years within the study area [26], stem discs were cut using a folding saw from the basal part of the shoot to allow the development of the longest possible ring-width series. After an initial visual inspection of stem cross sections, 4–5 stem discs sampled from each individual were discarded due to expression of extreme anomalous growth patterns, e.g., multisectoral eccentricity, lobed growth or high number of wedging rings. The remaining stem discs were air-dried, and the surface was prepared with a sharp razor blade and treated with wood stain to make the annual ring boundaries more clearly visible.

Particularly at the stem base, the arched ascending growth form of shoots of *Alnus alnobetula* cause eccentricity of the pith and elliptical-shaped annual increments. Therefore, radial growth was measured along four radii from the pith towards the cambium (Figure 1) including the longest and shortest radius, which were most frequently occurring at the lower (radius *a*) and upper (radius *b*) stem side, respectively. Additionally, radial growth along two radii (radius *c* and *d*) oriented orthogonal to the longest stem diameter were measured. Annual increments were measured along each radius to 1 µm precision using the LINTAB measuring system (Rinn, Heidelberg, Germany). A total of 188 radii were surveyed. Correct dating of ring-width time series (identifying missing or false rings) was checked by visually cross-dating all radii of one shoot, whereby special attention was given to the presence of extreme growth years, and finally using COFECHA software [49]. In a few cases it was necessary to insert a missing ring into the ring-width series (value of zero). Based on cross-dated ring-width time series measured from one stem disc, three radial combinations, i.e., averages of ≥2 radii, were calculated: mean of radii *a* and *b* (=*ab*), mean of radii *c* and *d* (=*cd*) and mean of all four radii (=*abcd*; Figure 1).

### 4.3. Statistical Analysis

Two different cross-dating methods were applied to quantify the agreement of the year-to-year ring-width changes among time series. Changes in ring width from year to year can be simplified to a binary variable (increase or decrease in ring width from one year to the next) and agreement can be quantified non-parametrically by counting the number of agreements and disagreements. Expressed as percentage of agreement, this is referred to as percentage of parallel variation [50] or “Gleichläufigkeit” (*Glk*; [51]). Agreement among ring-width values was also quantified parametrically using the product–moment correlation coefficient *r*, which in turn was adjusted for the amount of overlap between ring-width series using the standard *t*-statistic [52]. The value of *t* is defined as:$$t=\frac{r\cdot \sqrt{N-2}}{\sqrt{1-r^{2}}}$$
where *r* is the product–moment correlation coefficient and *N* is the amount of overlap between ring-width series.

A *t*_BP_-score of 3.5 is regarded as an acceptable agreement among ring-width series [50,52,53], and this *t*_BP_-score is also employed as a minimum for dating control in the software COFECHA. The value of *t* cannot be converted to a corresponding significance level, because autocorrelation is not considered [54] and cross-dating involves a high degree of multiplicity [55].

To determine the high-frequency (i.e., annual) agreement among ring-width series, age/size-related growth trends and other transient disturbance pulses had to be eliminated [55]. All ring-width series (i.e., ring-width series developed from single radii and all combinations of radii) were standardized to dimensionless radial growth indices by the ARSTAN program [56]. A conservative detrending method, i.e., a linear regression of negative slope or a negative exponential decline, was fitted to each ring-width series. Dimensionless radial growth indices were formed by dividing the observed ring-width value by the predicted ring-width value. Several statistics were then calculated for the standardized ring-width series. Mean sensitivity (MS) is a measure of the mean relative change between adjacent ring widths [33]. A high MS indicates that radial growth is highly responsive to the environment. The first-order autocorrelation (AC) assesses relationships with prior growth. The strength and confidence of ring-width series were estimated by calculating the signal-to-noise ratio (S/N-ratio) and the expressed population signal (EPS). The S/N-ratio is an expression of the strength of the observed common signal [57] and is defined as:S/N = N ∗ *r*/(1 − *r*) 
where *r* is the average correlation between radii and N is the number of radii. The mean correlation technique was applied to estimate the chronology signal strength (EPS), which quantifies the degree to which a given sample chronology reflects the hypothetically perfect chronology. Though a specific range of EPS values, which constitute acceptable statistical quality, cannot be given, Wigley et al. [57] suggested a threshold value of 0.85 as reasonable. Correlation matrices were calculated for the maximum period common to a specific radius and the mean inter-series correlation coefficient (Rbar) estimated according to formulae given in Briffa and Jones [58]. ARSTAN also calculates the common variance among ring-width series using principal component analysis [59]. Higher common variance accounted for by the first principal component (or eigenvector, EV) indicates a greater climatic influence on growth [33,58].

Student’s independent sample *t*-test was applied to determine significant differences among growth characteristics (shoot length, diameter and cambial age) of different stocks, and ring-width time series of single radii or combinations of radii. All the statistical analyses were performed using the STATISTICA software package (version 13.5.0.17; TIBCO Software Inc., Palo Alto, CA, USA).

## 5. Conclusions

We can summarize that at the alpine treeline, climate factors are the primary drivers of radial growth in the tall multi-stemmed shrub *Alnus alnobetula* leading to a high agreement among ring-width series developed from different radii, shoots and individuals, while differences in compressive and tensile forces and particularly variation in microsite conditions determine absolute growth rates and long-term growth trends. Thus, a high agreement of inter-annual radial growth contrasts with a high variability of growth rates, indicating that the research focus is also of central importance for the development of a suitable sampling design for dendroclimatological and ecological studies of tall multi-stemmed shrubs (*cf*. [21,60]).

## Figures and Tables

**Figure 1 plants-12-01708-f001:**
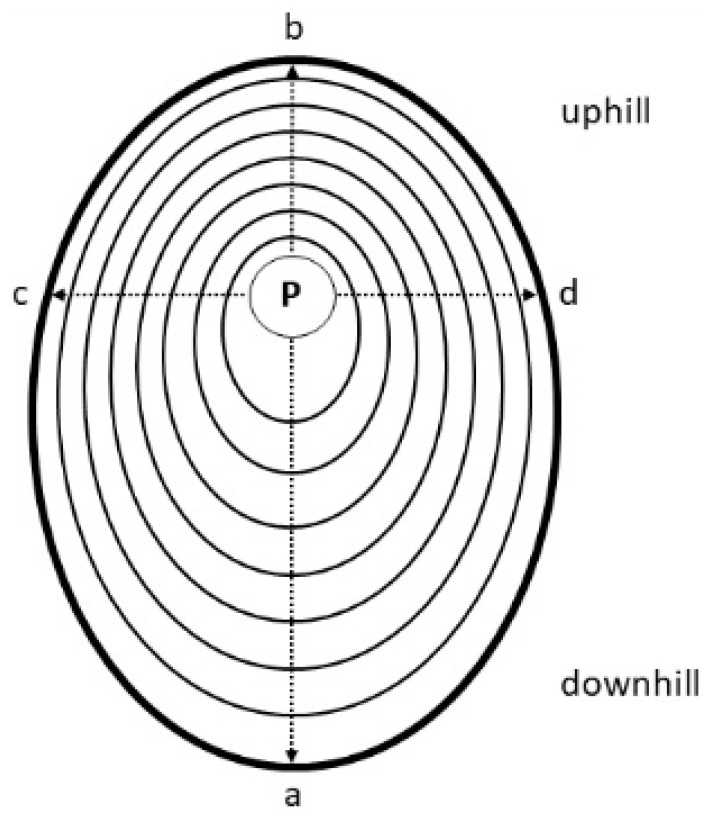
Schematic representation of measured radii. Radii *a* and *b* are oriented on the lower and upper side of the stem, respectively. Radii *c* and *d* are aligned perpendicular to these radii (*p* = pith).

**Figure 2 plants-12-01708-f002:**
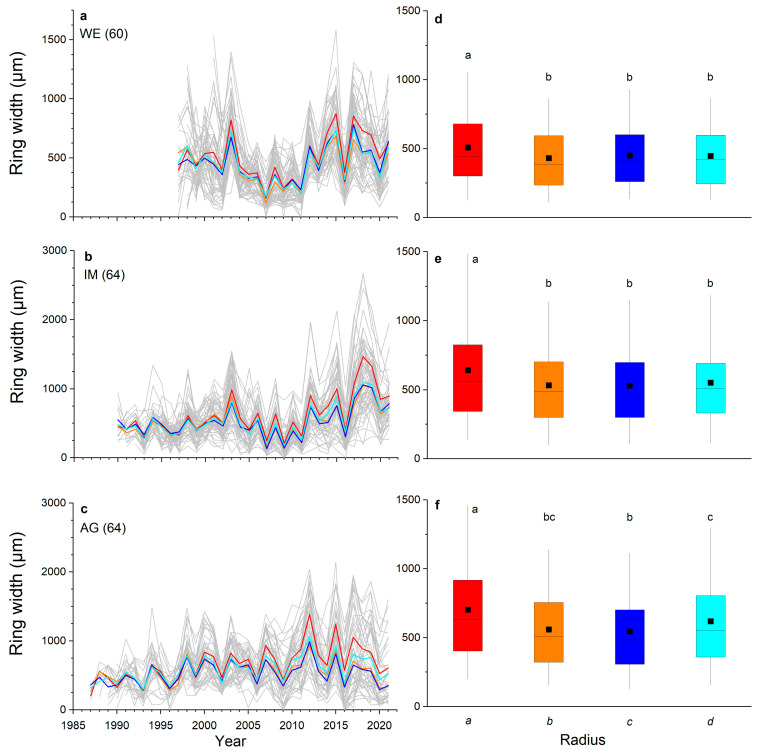
Ring-width series: (**a**–**c**) of all measured radii (grey lines; number of radii measured per stock are given in brackets) at a wind-exposed ridge (site WE), an intermediate site (IM) and a north-facing avalanche gully (site AG). Mean values of radii are shown in red (radius *a*), orange (radius *b*), dark blue (radius *c*) and light blue (radius *d*); (**d**–**f**) box-plots of ring width of all radii of each stock. The filled square and line indicate the mean and median ring width, respectively, the upper extent of the box indicates the 75th percentile, the lower extent indicates the 25th percentile and whiskers represent the 5th and 95th percentile. Statistically significant differences of mean values between radii are indicated by different letters (*p* ≤ 0.01; Student’s *t*-test).

**Figure 3 plants-12-01708-f003:**
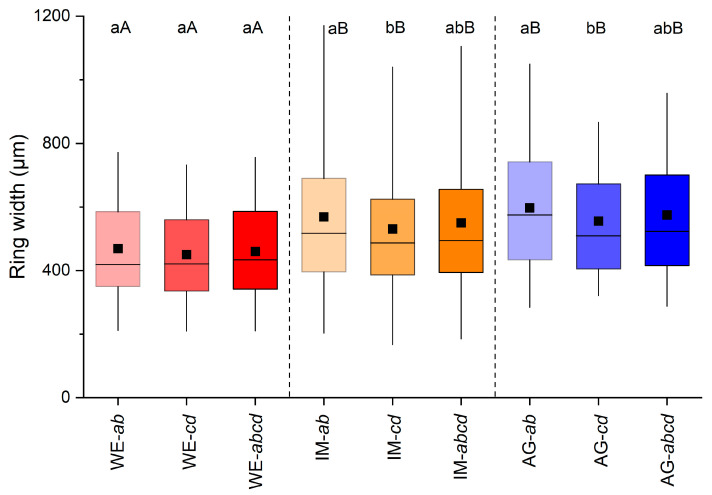
Box-plots showing variability in ring width among mean values of radii *a* and *b* (=*ab*), mean of radii measured perpendicular to these radii (=*cd*; cf. Figure 1) and all radii (=*abcd*) of all shoots taken from one stock at a wind-exposed ridge (site WE), an intermediate site (IM) and a north-facing avalanche gully (site AG). Statistically significant differences of mean values between and within stocks are indicated by different capital and small letters, respectively (*p* ≤ 0.05; Student’s *t*-test). For details on box-plots, see Figure 2.

**Figure 4 plants-12-01708-f004:**
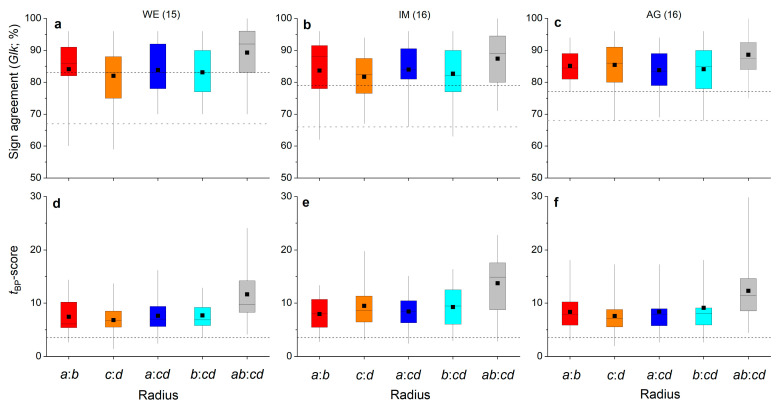
Box-plots showing variability in percentage of sign agreement ((**a**–**c**); *Glk* = Gleichläufigkeit; Eckstein and Bauch 1973) and the *t*_BP_-score ((**d**–**f**); Baillie and Pilcher 1990) among opposite radii (i.e., *a*:*b* and *c*:*d*), radii taken at the lower and upper side of the stem to the mean of radii measured perpendicular to these radii (i.e., *a*:*cd* and *b*:*cd*, respectively) and mean of radii *a* and *b* vs. mean of radii *c* and *d* (i.e., *ab*:*cd*) of all shoots taken from one stock at a wind-exposed ridge (site WE), an intermediate site (IM) and a north-facing avalanche gully (site AG). Number of shoots measured per stock are given in brackets. In (**a**–**c**), horizontal dashed black and grey lines indicate the significance level of *p* ≤ 0.001 and *p* ≤ 0.05, respectively. In (**d**–**f**), the horizontal dashed black line indicates a *t*-value of 3.5 (see Section 4 for details). For details on box-plots, see Figure 2.

**Figure 5 plants-12-01708-f005:**
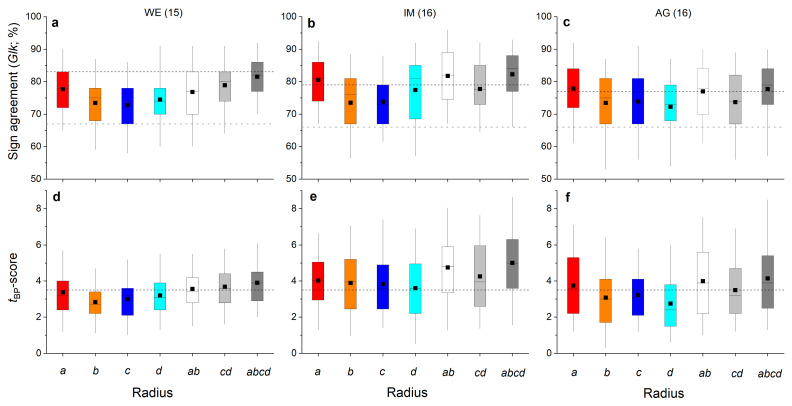
Box-plots showing variability in percentage of sign agreement ((**a**–**c**); *Glk* = Gleichläufigkeit; Eckstein and Bauch 1973) and *t*_BP_-score ((**d**–**f**); Baillie and Pilcher 1990) among single radii (i.e., *a*, *b*, *c* and *d*), means of opposite radii (i.e., *ab* and *cd*) and mean of all radii (i.e., *abcd*) of shoots taken from one stock at a wind-exposed ridge (site WE), an intermediate site (IM) and a north-facing avalanche gully (site AG). Number of shoots measured per stock are given in brackets. In (**a**–**c**), horizontal dashed black and grey lines indicate the significance level of *p* ≤ 0.001 and *p* ≤ 0.05, respectively. In (**d**–**f**), the horizontal dashed black line indicates a *t*-value of 3.5 (see Section 4 for details). For details on box-plots, see Figure 2.

**Figure 6 plants-12-01708-f006:**
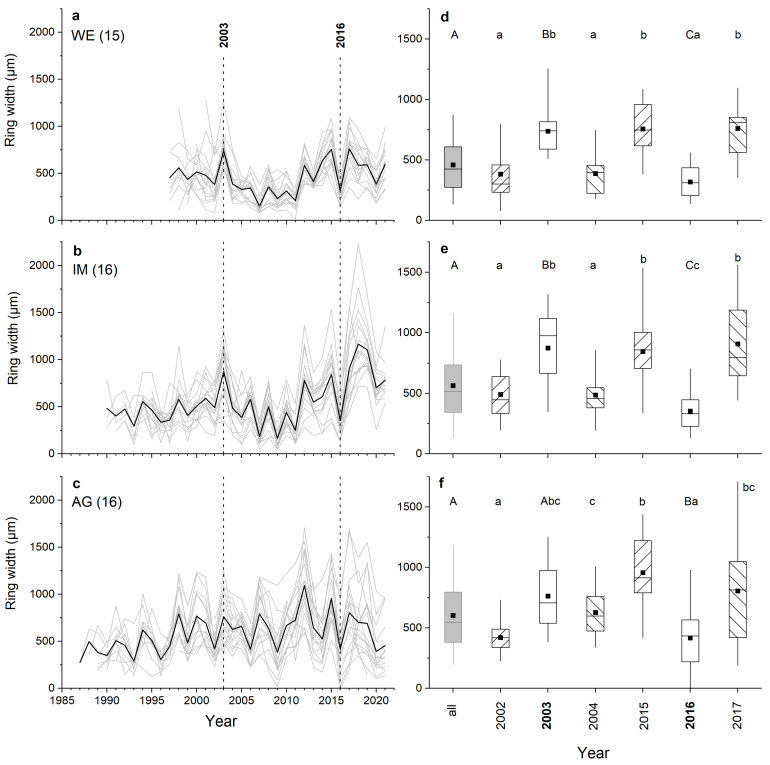
Ring-width series (**a**–**c**) of all radii (i.e., mean values of radii *a*, *b*, *c* and *d*) of all shoots taken from one individual (grey lines; number of shoots measured per stock are given in brackets) at a wind-exposed ridge (site WE), an intermediate site (IM) and a north-facing avalanche gully (site AG). Mean values of all shoots are given as black lines. Vertical dashed lines indicate exemplary extreme growth years: 2003 and 2016 show growth increase and decrease, respectively. Box-plots (**d**–**f**) showing ring-width variability in all years and in 2003 and 2016; the latter two cases include the previous and subsequent year. Statistically significant differences between (i) mean ring widths of all years and extreme growth years (2003, 2016) are indicated by upper case letters, and (ii) mean ring widths of extreme growth years and the preceding and subsequent year by lower case letters (*p* ≤ 0.001; Student’s *t*-test). For details on box-plots, see Figure 2.

**Figure 7 plants-12-01708-f007:**
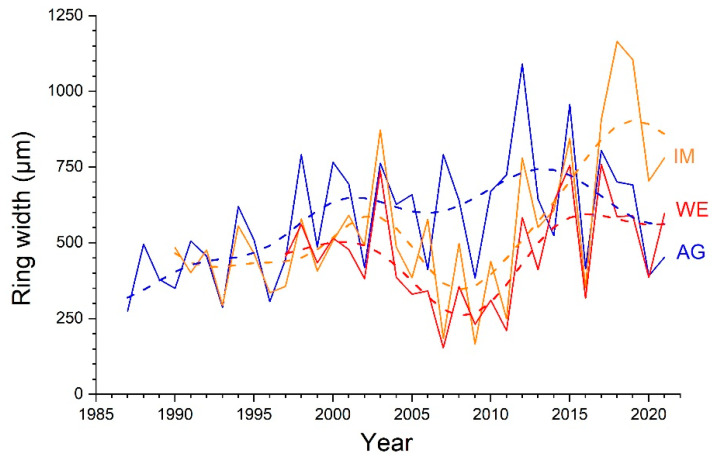
Long-term trend in ring-width series (mean of all radii, i.e., *abcd*) of all shoots of *Alnus alnobetula* sampled at a wind-exposed ridge (site WE), an intermediate site (IM) and a north-facing avalanche gully (site AG). Long-term trends (dashed lines) were calculated by fast Fourier transform low-pass filter (3 yrs).

**Table 1 plants-12-01708-t001:** Characteristics of selected *Alnus alnobetula* individuals (RW = ring width; SD = standard deviation; ShA = shoot age; ShD = shoot diameter; ShL = shoot length; sites: WE = wind-exposed ridge; IM = intermediate site; AG = avalanche gully). Statistically significant differences of mean values among individuals (=stocks) are indicated by different letters (*p* ≤ 0.01; Student’s *t*-test).

Site	Shoots ^1^ (*n*)	ShL (m) Mean ± SD	ShD (cm) Mean ± SD	ShA ^2^ (yrs) Min/Max/Mean ± SD	RW ^3^ (µm) Mean ± SD
WE	15	1.6 ± 0.2 ^a^	2.7 ± 0.3 ^a^	19/25/23.5 ± 1.8 ^a^	462 ± 59 ^a^
IM	16	2.5 ± 0.1 ^b^	3.8 ± 0.6 ^b^	23/32/28.6 ± 2.6 ^b^	566 ± 107 ^b^
AG	16	2.5 ± 0.3 ^b^	4.0 ± 0.7 ^b^	19/35/29.2 ± 5.5 ^b^	618 ± 134 ^b^

^1^ Number of shoots taken from one individual (total number of shoots varied between 30–40 for each stock). ^2^ Mean cambial age of shoots (stem discs were taken close to the stem base). ^3^ Mean of all four radii measured in all shoots belonging to one stock.

**Table 2 plants-12-01708-t002:** Statistics of ring-width series measured along different radii (*a*–*d*) and mean values of combinations of radii (*ab*, *cd*, *abcd*) of all shoots taken from one stock at a wind-exposed ridge (site WE), an intermediate site (IM) and a north-facing avalanche gully (site AG). Statistics are calculated based on detrended ring-width series for the common interval 2001–2021 (AC = autocorrelation; EPS = expressed population signal; EV = variance 1st eigenvector; MS = mean sensitivity; Rbar = mean inter-series correlation among radii).

Site	Radius	MS (%)	AC	EPS	S/N-Ratio	Rbar	EV (%)
WE	*a*	41.9	0.255	0.961	24.9	0.657	68.8
	*b*	41.5	0.109	0.950	19.0	0.593	62.6
	*c*	42.2	0.112	0.943	16.5	0.559	60.2
	*d*	41.9	0.124	0.951	19.5	0.600	64.6
	*ab*	40.4	0.226	0.964	27.0	0.675	70.2
	*cd*	40.7	0.117	0.956	21.8	0.627	66.4
	*abcd*	40.5	0.171	0.966	28.1	0.684	71.2
IM	*a*	47.4	0.430	0.966	28.8	0.643	68.9
	*b*	48.9	0.326	0.959	23.3	0.593	63.5
	*c*	47.9	0.349	0.954	20.5	0.562	62.5
	*d*	45.5	0.333	0.970	32.3	0.669	69.4
	*ab*	47.7	0.389	0.970	32.4	0.669	70.7
	*cd*	45.4	0.332	0.971	34.0	0.680	70.7
	*abcd*	45.8	0.365	0.975	38.7	0.708	73.5
AG	*a*	41.2	0.208	0.919	11.3	0.465	52.6
	*b*	41.6	−0.058	0.904	9.4	0.419	49.2
	*c*	40.0	−0.026	0.894	8.4	0.392	47.2
	*d*	36.7	0.074	0.861	6.2	0.324	41.5
	*ab*	41.2	0.099	0.928	12.9	0.497	55.4
	*cd*	38.0	0.019	0.898	8.8	0.405	48.2
	*abcd*	39.4	0.072	0.926	12.5	0.489	55.4

**Table 3 plants-12-01708-t003:** Percentage of sign agreement (*Glk*), statistical significance and *t*_BP_-scores (given after) calculated among ring-width series of single radii (radius *a*, *b*, *c*, *d*), mean values of combinations of radii (*a* and *b* = *ab; c* and *d* = *cd*) and all radii (=*abcd*) among stocks sampled at a wind-exposed ridge (site WE), an intermediate site (IM) and a north-facing avalanche gully (site AG). Length of time series are given in brackets. * = *p* ≤ 0.05; ** = *p* ≤ 0.01; *** = *p* ≤ 0.001.

	Radius *a*			Radius *b*			Radius *c*		Radius *d*				Mean *ab*			Mean *cd*			Mean *cd*		
	WE	IM	AG	WE	IM	AG	WE	IM	AG	WE	IM	AG	WE	IM	AG	WE	IM	AG	WE	IM	AG
WE (25)	-	96 ***/8.4	71 */2.3	-	83 ***/7.2	75 **/1.9	-	88 ***/6.7	71 */2.5	-	83 ***/7	81 ***/2.9	-	92 ***/8.2	75 **/2.2	-	88 ***/7	73 */2.7	-	88 ***/7.6	71 */2.4
IM (32)		-	65/3.6		-	68 */3.6		-	68 */3.4		-	66 */3.9		-	68 */3.6		-	66 */3.6		-	68 */3.6
AG (35)			-			-			-			-			-			-			-

## Data Availability

The ring-width data presented in this study are openly available in Zenodo at 10.5281/zenodo.7708713.

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
