# Peer review of "Climate Overrides the Influence of Microsite Conditions on Radial Growth of the Tall Multi-Stemmed Shrub Alnus alnobetula at Treeline"

_plants, 2023, doi:10.3390/plants12081708_

Round 1

Reviewer 1 Report

This research is very interesting. But some organization of the manuscript was not accepted such that the hepothesis was shown in the Abstract but no responses to it. Another typical special arrangement is that the discussion was included in the Conclusion. I suggest to rearrange and improve the manuscript to be more feasible for readers.

Author Response

We thank the reviewer for the positive evaluation and helpful comments. We agree that parts of the discussion were included in the conclusion section and have re-arranged this as suggested. We have left the abstract unchanged, because of the given restrictions regarding the number of words (200 maximum), and because we have mentioned the aims and main results of our study, albeit in a very condensed way.

Reviewer 2 Report

The study is well designed and presented. I have only few small notes.

·       It would be good to add a photo of the species under study. Also it would be good to clarify, how far from each other could be individual stems of the same sample stock – if they are growing evidently from the same center ±20-40 cm, or they are spread up to several meters from each other.

·       It would be good also to add weather characteristics of the period under study, taking into account that the climate dependence of growth is discussed (in particular, extreme growth years), but no weather time series is presented.

·       I am not sure that it is correct to call AG site as specially exposed to high disturbance regime – it is question, what is higher disturbance: high snow cover in AG or strong wind in WE. Or you mean that an avalanche could happen during the grows of the studied stems?

·       Among statistical comparisons in this study (within one stem, within one shrub and among plots) one is missing: between different shrubs in the same plot (although in L.372-374 authors recommend to measure ~10 individual shrubs in homogenous conditions). It could be recommended to do in future.

Some other small notes are in the attachment

Author Response

The study is well designed and presented. I have only few small notes.

 It would be good to add a photo of the species under study. Also it would be good to clarify, how far from each other could be individual stems of the same sample stock – if they are growing evidently from the same center ±20-40 cm, or they are spread up to several meters from each other.

We thank the reviewer for the very detailed suggestions, which has definitely helped to improve our manuscript. As suggested we added a photograph of the species under study as Supplementary Material (Figure S2). We have also included a sentence in MM section describing the maximum distance of the sampled shoots from the center of the stock.

It would be good also to add weather characteristics of the period under study, taking into account that the climate dependence of growth is discussed (in particular, extreme growth years), but no weather time series is presented.

Yes, we agree and added a graph showing records of air temperature and precipitation at Mt. Patscherkofel during the study period (Supplementary Figure S1).

I am not sure that it is correct to call AG site as specially exposed to high disturbance regime – it is question, what is higher disturbance: high snow cover in AG or strong wind in WE. Or you mean that an avalanche could happen during the grows of the studied stems?

The AG site was considered to be specially exposed to a high disturbance regime, because we found the highest variability in cambial age of sampled shoots in this individual, ranging from 19 to 35 years (cf. Table 1). Repeated occurrences of avalanches have most likely caused successive sprouting at this site.

Among statistical comparisons in this study (within one stem, within one shrub and among plots) one is missing: between different shrubs in the same plot (although in l.372-374 authors recommend to measure ~10 individual shrubs in homogenous conditions). it could be recommended to do in future.

We agree and have also added a sentence in the discussion section.

Some other small notes are in the attachment

These comments were very helpful. We have taken them all into account in the revised manuscript and have also added relevant references if necessary.

Round 2

Reviewer 1 Report

The manuscript is attecptable now.